# Strengthening National Regulatory Authorities in Africa: A Critical Step Towards Enhancing Local Manufacturing of Vaccines and Health Products

**DOI:** 10.3390/vaccines13060646

**Published:** 2025-06-16

**Authors:** Alemayehu Duga, Nebiyu Dereje, Mosoka Papa Fallah, Tedi Angasa, Abebe Genetu Bayih, Edinam Agbenu, Ngashi Ngongo, Raji Tajudeen, Jean Kaseya

**Affiliations:** 1Africa Centres for Disease Control and Prevention, Addis Ababa 3243, Ethiopia; alemayehud@africacdc.org (A.D.); fallahm@africacdc.org (M.P.F.); tedia@africacdc.org (T.A.); abebeg@africacdc.org (A.G.B.); ngongon@africacdc.org (N.N.); tajudeenr@africacdc.org (R.T.); kaseyaj@africacdc.org (J.K.); 2Intercountry Support Team, Regional Office for Africa, World Health Organization, Ouagadougou 100701, Burkina Faso; agbenue@who.int

**Keywords:** regulatory maturity, counterfeit medicines, local manufacturing, pharmacovigilance system, emergency preparedness, safety and efficacy

## Abstract

The World Health Organization (WHO) Global Benchmarking Tool (GBT) classifies regulatory systems into four maturity levels, with Maturity Level 3 (ML3) signifying a stable and effective regulatory environment. As of January 2025, eight African nations—Egypt, Ghana, Nigeria, Rwanda, Senegal, South Africa, Tanzania, and Zimbabwe—have attained ML3 status, marking a significant milestone in the continent’s regulatory landscape. Achieving ML3 confers critical benefits, including reducing substandard and falsified medicines, which enhances public health safety and fosters trust in healthcare systems. This progress encourages local manufacturing, diminishing reliance on imported medicines and promoting economic development. Furthermore, ML3 NRAs are better equipped to address public health emergencies, enabling swift approvals for vaccines and therapeutics while upholding safety standards. Nonetheless, challenges persist, including fragmented regulatory systems, the prevalence of counterfeit medicines, and limited resources. Overcoming these hurdles necessitates enhanced organizational capacity, investments in training, and the promotion of collaboration among NRAs. There is an urgent call for greater political commitment and resource allocation to strengthen regulatory systems across Africa. Achieving and maintaining ML3 status is essential for enhancing medicine regulation, supporting local manufacturing, and improving public health outcomes across the continent. While progress has been made, sustained efforts are crucial to tackling existing challenges and harnessing the full potential of advanced regulatory frameworks.

## 1. Introduction

Africa, with a sizable population, has been affected by recurrent disease outbreaks but has only limited resilience in its health systems to mitigate the impacts of these outbreaks [1]. One of the challenges to responding to these outbreaks is the limited access to medical products such as vaccines, therapeutics, and diagnostics. The role of national regulatory authorities (NRAs) in contributing to access to safe and effective medical products, including vaccines, is indispensable. However, most NRAs in Africa lack the required capacities and capabilities to optimally support the supply and local manufacturing of vaccines and other health products [2]. NRAs are crucial in ensuring health products’ safety, efficacy, and quality [2]. Strong NRAs are vital for protecting public health, fostering innovations and local manufacturing of health products, and increasing access to essential medicines, particularly during responses to public health emergencies [2].

To continuously evaluate the capacity and capability of NRAs, the World Health Organization (WHO) has established a regulatory benchmarking framework based on over 250 indicators. The Global Benchmarking Tool (GBT) evaluates NRA systems and categorizes them into four maturity levels (ML 1–4). Achieving Regulatory Maturity Level 3 (ML3) under this framework indicates that an NRA has a stable, well-functioning regulatory system. Maturity Level 4, the highest level, represents an advanced regulatory system dedicated to continuous improvement and innovation [3]. This framework establishes benchmarks for regulatory excellence and serves as a catalyst for strengthening healthcare systems worldwide. As countries strive for regulatory maturity, the implications extend beyond compliance, fostering trust and enhancing public health outcomes.

The landscape of the medicine regulatory system in Africa is undergoing a significant transformation, driven by the increasing number of NRAs achieving ML3. This milestone is a fundamental shift required for the continent’s New Public Health Order that prioritizes local manufacturing of health products to ensure continental health security [4] with minimal external dependency. Notably, the huge disparity in access to medical products such as vaccines during major public health emergencies, such as Ebola and COVID-19, has remained a challenge in Africa. Even now, in the ongoing outbreaks of mpox, Marburg disease, and Sudan Ebola disease in multiple parts of Africa, access to vaccines has remained a major challenge, reaffirming the need for local manufacturing of vaccines, particularly those targeting epidemic-prone diseases. This underscores the need for strengthened national regulatory authorities to accelerate the timely regulatory approval of the vaccines, enhancing investments for local vaccine production.

## 2. The Current Maturity Level Status of NRAs in Africa

As of January 2025, eight African countries (Egypt, Ghana, Nigeria, Rwanda, Senegal, South Africa, Tanzania, and Zimbabwe) have successfully achieved ML3 status for medicines and vaccines, demonstrating significant progress in strengthening the regulatory system (Table 1). South Africa and Egypt achieved ML3 for vaccine production. At the same time, Egypt stands out as the only country in Africa to attain this important milestone for both medicines and vaccines, showcasing its leadership and commitment to regulatory system advancement [5]. This progress has been made possible through a combination of domestic government investment, donor funding, and technical and financial support from multilateral agencies such as WHO, Gavi, the Vaccine Alliance, and other funding partners. Although this is encouraging, most African countries affected by recurrent public health emergencies and needing essential medical products to control them must work towards having robust national regulatory authorities.

## 3. Key Impacts of Advanced Regulatory Maturity

ML3 also promotes local manufacturing of vaccines and health products through investment and innovation by creating a stable regulatory environment, which reduces reliance on imported medicines. In Africa, 95% of medicines and 99% of vaccines are imported, leaving healthcare systems vulnerable to supply chain disruptions and price volatility [6]. This reliance on external supply sources presents various challenges, including restricted access to essential medicines during global health crises, as seen during the COVID-19 pandemic [7]. A robust regulatory framework attracts investments, encourages technological advancements, and generates job opportunities, stimulating economic development and alleviating poverty. The target of the African Union and Africa Centres for Disease Control and Prevention (Africa CDC) is to ensure local manufacturing of 60% of the vaccines, diagnostics, and medicines used in Africa by 2040. This is expected to create a market size of USD 50 billion in Africa [8].

The mpox and Marburg virus disease outbreaks underscored the need for flexible and effective regulatory systems. NRAs operating at ML3 are particularly well suited to address these challenges by accelerating the approval and distribution of vaccines and therapeutics, upholding safety and efficacy standards [9]. ML3 systems enhance health system resilience by streamlining processes for emergency use authorizations, facilitating the rapid deployment of vaccines and medical products, and maintaining rigorous oversight throughout crises. For instance, during the COVID-19 pandemic, countries with advanced regulatory systems were vital in facilitating the global distribution of vaccines under emergency use listings [10]. ML3 supports establishing pre-approved contingency plans and partnerships with manufacturers, ensuring rapid production scalability during emergencies. This proactive approach is crucial for maintaining a steady supply of essential medicines and vaccines, particularly in the face of unexpected public health challenges. It also enables robust pharmacovigilance systems to monitor the safety of newly deployed products, providing real-time data to inform public health decisions. This ongoing monitoring is essential for identifying potential adverse effects and ensuring that the benefits of medical products outweigh the risks, thereby maintaining public confidence in healthcare interventions.

In addition to contributing to local manufacturing of vaccines, strengthening NRAs offers significant benefits in reducing substandard and falsified medicines that challenge the health system in Africa. More than 10% of medicines in Africa are substandard or falsified—claiming lives, resulting in economic loss, and contributing to drug resistance [11]. This underscores the urgent need to enhance the regulatory and quality assurance processes of NRAs to protect public health by ensuring that medicines meet strict safety and efficacy standards, fostering trust in healthcare systems and improving patient safety. Counterfeit and substandard medicinal products, particularly life-saving medicines, pose a significant threat to public health. Incidents like the deaths of 66 children in Gambia in 2022 from contaminated cough syrup underscore the seriousness of this issue [12]. Furthermore, 122,000 children in sub-Saharan Africa die each year due to poor-quality antimalarials [13]. Africa has the highest counterfeit and substandard medical product rate (18.7%) globally. The global market for these products is substantial, estimated to be between USD 65 billion and USD 200 billion annually [13]. Weak regulatory environments exacerbate this problem, eroding public trust and hindering efforts to improve regulatory standards. The pursuit of achieving ML3 goes beyond mere regulatory compliance; it embodies a deeper dedication to health equity and socioeconomic development. As African countries work towards regulatory excellence, they not only safeguard their citizens but also establish the foundation for a healthier future.

## 4. Challenges in Implementing ML3

Despite the notable progress and advantages of ML3 implementation in Africa, several challenges impede its expanded adoption and effectiveness. A primary challenge is the fragmented and underdeveloped regulatory systems across African countries. This fragmentation causes inefficiencies and delays in regulatory approvals, ultimately restricting access to essential medicines [14]. Moreover, weak pharmacovigilance systems across the continent pose a critical concern. The limited number of Individual Case Safety Reports (ICSRs) submitted from Africa to VigiBase, constituting less than 1% of global reports [15], underscores significant under-reporting, which hinders effective drug and vaccine safety monitoring and undermines public confidence in the safety and efficacy of medical products.

A lack of political commitment and financial resources often hampers the implementation of ML3 in many African countries. This insufficient support leads to inadequate funding for regulatory activities and slow progress in harmonizing regulations. In addition, infrastructural and technical challenges further obstruct the implementation of ML3 in Africa. These challenges include inadequate pharmaceutical manufacturing infrastructure, lack of skilled and fully dedicated personnel, weak and often manual and fragmented data systems and limited use of innovative technologies for operational efficiency [14]. These limitations can delay the development and implementation of robust regulatory systems.

To enhance the effectiveness and improvement of regulatory systems across Africa, it is crucial to promote inter-country and stakeholder collaboration. This strategy addresses the persistent challenges associated with fragmented regulatory frameworks and the limitations of underdeveloped systems that impede progress. By establishing regional regulatory networks, nations can exchange best practices, resources, and expertise, thereby optimizing regulatory processes across borders. Such integration facilitates the harmonization of regulations and the establishment of mutual recognition agreements, essential for accelerating access to safe medicines and vaccines. Furthermore, these partnerships allow for pooled resources in training regulatory personnel, sharing data, and performing joint evaluations of pharmaceuticals and vaccines. By leveraging the collective strengths of member states, regulatory authorities can create a more resilient and efficient system that benefits all stakeholders involved.

In parallel, the adoption of innovative technologies is vital for the modernization of regulatory practices. The integration of artificial intelligence (AI) and machine learning can streamline various regulatory processes, ranging from application assessments to post-market surveillance. These technologies can significantly enhance the identification of potential safety concerns more swiftly and with greater precision than traditional methodologies, thereby bolstering pharmacovigilance efforts. Additionally, AI-driven data analysis capabilities empower regulators to make informed decisions in real time, particularly crucial during public health emergencies. Utilizing digital platforms for the reporting and monitoring of ICSRs can markedly improve data collection, addressing the prevalent under-reporting issue in numerous African nations.

Investment in technological infrastructure is imperative for regulatory agencies to adapt to the dynamic landscape of medicine distribution and public health. Implementing electronic systems for document management and communication can effectively minimize approval delays and enhance operational transparency. This modernization not only augments regulatory efficiency but also strengthens public confidence by making processes more transparent and accountable.

Lastly, sustained engagement with stakeholders, including pharmaceutical manufacturers, healthcare practitioners, and civil society, is essential for crafting policies that accurately reflect the population’s needs. By fostering an inclusive approach to regulatory development, authorities can ensure that regulations are both effective and equitable, and accessible to all demographic segments. Increased financial commitments from governments and international organizations will support these initiatives, fostering the growth of robust pharmaceutical manufacturing capabilities on the continent while reducing reliance on imports. This, in turn, will ensure a reliable supply of essential medicines during public health crises.

## 5. Call to Action

As we envision sustainable and diversified local manufacturing of vaccines and health products, strengthening national regulatory authorities in Africa is critical. Building organizational capacity and training regulatory personnel are crucial to achieving and maintaining regulatory ML3 status. We call upon the high-level leadership of African countries and stakeholders to prioritize strengthening NRAs in Africa through innovative technologies that streamline operations and improve efficiency, allocating resources, and developing or adopting national policies, guidelines, and frameworks addressing the needs of NRAs. This mechanism should promote political ownership, sustainable domestic financing, and policy integration across the health, industry, and education sectors. It must also ensure regular peer reviews, performance benchmarking, and joint capacity-building programs supported by South–South and North–South partnerships. Member states should embed NRA development into national health security and industrialization plans, ensure legal autonomy for regulators, and foster regulatory science hubs linked to academia. In parallel, digital innovation, pooled procurement mechanisms, and shared regulatory functions—such as joint inspections and product assessments—should be institutionalized to optimize resources. These collective actions will empower NRAs to achieve WHO maturity levels, ensure product quality, and support the continent’s vision for pharmaceutical sovereignty. Moreover, unified efforts of national and international organizations, such as Africa CDC and the African Medicine Agency (AMA), are required to strengthen the NRAs in Africa to ensure diversified and sustainable manufacturing of vaccines and health products.

## Figures and Tables

**Table 1 vaccines-13-00646-t001:** List of national regulatory authorities (NRAs) operating at Maturity Level 3 (ML3).

Category	Country	NRA	Maturity Level	Year of Announcement
Non-Producing Vaccines	Ghana	Food and Drugs Authority (FDA)	ML3	2020
	Nigeria	National Agency for Food and Drug Administration (NAFDAC)	ML3	2022
	Rwanda	Food and Drugs Authority (Rwanda FDA)	ML3	2024
	Sénégal	Agence sénégalaise de Réglementation Pharmaceutique	ML3	2024
	Tanzania	Tanzania Medicines and Medical Devices Authority (TMDA)	ML3	2018
	Zimbabwe	Medicines Control Authority of Zimbabwe (MCAZ)	ML3	2024
Producing Vaccines	South Africa	South African Health Products Regulatory Authority (SAHPRA)	ML3	2022
Medicines and Vaccines	Egypt	Egyptian Drug Authority (EDA)	ML3	2022/2024

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
