# Peer review of "Strengthening National Regulatory Authorities in Africa: A Critical Step Towards Enhancing Local Manufacturing of Vaccines and Health Products"

_vaccines, 2025, doi:10.3390/vaccines13060646_

Round 1
Reviewer 1 Report
Comments and Suggestions for Authors
The submitted Commentary provides a brief update on the status, and generalized need for improvement, of National Regulatory Authorities (NRA) in Africa. While limitations are presented, remedies (or suggested remedies) are limited. Irrespective of this, the submission is warranted and its dissemination important.
I have one singular comment, in that sentence 1 is needs attention as it does not make immediate sense - perhaps rephrase to "Africa, with a sizable population, has been affected by recurrent disease outbreaks besides but has only limited resilience of health systems to mitigate the impacts of these outbreaks"
Author Response
The submitted Commentary provides a brief update on the status, and generalized need for improvement, of National Regulatory Authorities (NRA) in Africa. While limitations are presented, remedies (or suggested remedies) are limited. Irrespective of this, the submission is warranted and its dissemination important.
I have one singular comment, in that sentence 1 is needs attention as it does not make immediate sense - perhaps rephrase to "Africa, with a sizable population, has been affected by recurrent disease outbreaks besides but has only limited resilience of health systems to mitigate the impacts of these outbreaks"
Response: We thank the reviewer for this comment. We have corrected the sentence as suggested.
Reviewer 2 Report
Comments and Suggestions for Authors
Authors underline the importance of NRAs in the development of medicine and vaccine manufacturing ecosystem in Africa. They discuss the status and challenges faced by these institutions. This commentary is highly appreciated, however there are few points that require attention from the authors. Pls see below:
1. Strengthen the commentary by providing statistics wherever possible (although reference have been given, it is crucial to make a point). This is highly recommended.
2. Can the authors provide some data on the number of NRAs that are currently at ML1 and ML2.
3. Can the authors comment on the fundings that supported the establishment of ML3 efforts (table 1). Are there any estimations (cost, manpower, time) for transforming ML2 to either ML3 or ML4 for vaccine, medicine production?
4. Are there any countries that have dedicated (direct or indirect) budget allocation towards this effort and how much?
5. How many ML3 and ML4 NRAs are required to fulfill the demands of Africa. Are there any modeling or estimations studies regarding this.
6. For establishing additional ML3 and ML4 NRAs, do countries have necessary technical know-how and industrial capabilities to foster local manufacturing? Pls comment with any on-going or prospective programs that are trying to achieve this.
7. Pls enlist any collaborative or partnerships between countries at business to business or government to government level to build manufacturing capabilities and strengthen NRAs.
8. Mention any efforts from countries that have a mandate to nourish the NRAs as state policy.
9. Are there any efforts (funding, know-how or in any form) from non-African countries to assist NRA.
10. The authors have commented on the challenges for development and set up of NRAs at several levels. They have called upon the leadership of African countries to improve on these aspects, so it will be great to expand on this section. Pls elaborate on (a high-level) a mechanism to address these challenges. Make recommendations and provide ideas and solutions. (dedicate 3-4 paragraphs for this section).
Author Response
- Strengthen the commentary by providing statistics wherever possible (although references have been given, it is crucial to make a point). This is highly recommended.
Response: Thank you. We have provided statistics whenever possible.
- Can the authors provide some data on the number of NRAs that are currently at ML1 and ML2.
Response: The specific numbers of Maturity Levels 1 and 2 (ML1 and ML2) NRAS in Africa are not publicly available. However, it's understood that the majority of African countries are still working towards achieving ML3 status. Maturity Level 1 signifies that some elements of a regulatory system exist, while Level 2 indicates an evolving system that partially performs essential regulatory functions.
- Can the authors comment on the fundings that supported the establishment of ML3 efforts (table 1). Are there any estimations (cost, manpower, time) for transforming ML2 to either ML3 or ML4 for vaccine, medicine production?
Response: Thank you. We have added the following sentence to highlight this.
The establishment and advancement of NRAs to Maturity Level 3 (ML3), as presented in Table 1, have been primarily supported through a mix of government funding, donor funding, and technical and financial assistance from multilateral agencies such as the World Health Organization (WHO), Gavi, the Vaccine Alliance and other donor organizations. However, the actual cost for the establishment varies across the country’s contexts.
- Are there any countries that have dedicated (direct or indirect) budget allocation towards this effort and how much?
Response: As this article is a commentary based on available data, this information is not readily available and not included in the article.
- How many ML3 and ML4 NRAs are required to fulfill the demands of Africa. Are there any modeling or estimations studies regarding this.
Response: Currently, no universally agreed-upon model or estimation that defines the exact number of ML3 or ML4 NRAs required to meet Africa’s comprehensive regulatory needs. However, establishing a network of ML3/ML4 NRAs, well-advanced regulatory regional and continental blocks across the continent to serve as reference authorities will help achieve the goal.
- For establishing additional ML3 and ML4 NRAs, do countries have necessary technical know-how and industrial capabilities to foster local manufacturing? Pls comment with any on-going or prospective programs that are trying to achieve this.
Response: Thank you for this comment. We have addressed as follows.
Most African countries still lack the technical and industrial capacity needed for ML3 and ML4 NRAs, though progress is underway. Initiatives like the African Medicines Agency (AMA), WHO’s Global Benchmarking Tool, and the Platform for Harmonised African Health Manufacturing (PHAHM)are supporting regulatory strengthening. Programs such as MAV+ are also promoting local manufacturing through capacity building and technology transfer. With sustained investment and policy alignment, more countries can reach higher regulatory maturity.
- Pls enlist any collaborative or partnerships between countries at business to business or government to government level to build manufacturing capabilities and strengthen NRAs.
Response: One of the most significant collaborations is the AUDA-NEPAD AMRH initiative, alongside Africa CDC’s PHAHM, both supporting regulatory harmonization and capacity building. These efforts promote technology transfer and sustainable local manufacturing. B2B and G2G partnerships like BioNTech-Rwanda and Biovac-Senegal further strengthen this agenda. Additional collaborations are detailed within the manuscript. This has been well covered between line #75-79
- Mention any efforts from countries that have a mandate to nourish the NRAs as state policy.
Response: Countries like Ghana, Rwanda, South Africa, Nigeria, and Egypt have integrated NRA strengthening into national policies. Their governments provide direct support through reforms, funding, and strategic alignment with health and industrial goals. Ghana’s FDA and Nigeria’s NAFDAC are notable examples with WHO-recognized progress. These efforts reflect a growing recognition of NRAs as critical to health system resilience and local manufacturing.
- Are there any efforts (funding, know-how or in any form) from non-African countries to assist NRA.
Response: Initiatives such as USAID, GIZ, and the EU's MAV+ program enhance regulatory capacity and harmonization. These efforts aim to strengthen NRAs and support sustainable local manufacturing.
- The authors have commented on the challenges for development and set up of NRAs at several levels. They have called upon the leadership of African countries to improve on these aspects, so it will be great to expand on this section. Pls elaborate on (a high-level) a mechanism to address these challenges. Make recommendations and provide ideas and solutions. (dedicate 3-4 paragraphs for this section).
Response: Thank you for your comment; we appreciate your insights. We have included a summarized call to action reflecting your input. However, due to the manuscript being a commentary, we kept it to a single paragraph to maintain focus and deliver a clear and unambiguous message. We added the following in the last paragraph: This approach should encourage political ownership, sustainable domestic financing, and policy integration across health, industry, and education sectors. It must also facilitate regular peer reviews, performance benchmarking, and collaborative capacity-building programs, supported by both South-South and North-South partnerships. Member states should integrate NRA development into national health security and industrialization strategies, ensure legal autonomy for regulators, and promote regulatory science hubs in collaboration with academia. Concurrently, digital innovation, pooled procurement strategies, and shared regulatory functions—such as joint inspections and product evaluations—should be established to enhance resource optimization. These collective efforts will enable NRAs to reach WHO maturity levels, guarantee product quality, and advance the continent’s vision for pharmaceutical sovereignty.
Reviewer 3 Report
Comments and Suggestions for Authors
Commentary: Strengthening National Regulatory Authorities in Africa: A Critical Step Towards Enhancing Local Manufacturing of Vaccines and Health Products
A clearly stated purpose of the document would be helpful, which could be:
- the need for investment in strengthened national regulatory authorities to accelerate timely regulatory approval AND LOCAL MANUFACTURING of SAFE, EFFECTIVE vaccines (to achieve regulatory ML3 status)
Add headings and subheadings may give more structure to the document.
Some suggestions:
The issue – safe, efficacious medical products needed in Africa
The role of National Regulatory Authorities (beginning line 15)
National Regulatory Authority levels of expertise – WHO benchmarking framework (line 23)
The regulatory system in Africa (line 30)
Dependence on imported medical products and vaccines (line 55)
Addressing substandard and falsified medicines (line 77)
Challenges for strong regulation (line 92)
Need for investment in national regulatory authorities (line 111)
Use of ‘countermeasure’ throughout – not a term so commonly used in health - replace with 'medical products'
Line 12: Africa, with a sizable population, has been affected by recurrent disease outbreaks and yet its health systems have [besides its] limited resilience [of health systems] to mitigate the impacts of these outbreaks.
One of the challenges to responding to these outbreaks is the limited access to medical Countermeasures/PRODUCTS such as vaccines, therapeutics and diagnostics.
LINE 15: ..The role of National Regulatory Authorities (NRAs) in contributing to/ENABLING access to SAFE AND EFFECTIVE medical countermeasures/PRODUCTS, including vaccines, is indispensable/CRITICAL. However, most NRAs in Africa lack the required capacities and capabilities to optimally support the SUPPLY AND local manufacturing of ..
LINE 22: ..during responseS to public health
LINE 23: To continuously evaluate the level of CAPACITY AND CAPABILITY OF NRAs, the World Health Organization (WHO)
LINE 32: is a fundamental shift in/REQUIRED FOR the continent's New Public Health Order that
LINE 34: Notably, the huge disparity in access to medical countermeasures/PRODUCTS such as vaccines during public major public health emergencies, [such] as OCCUR DURING SPREADS OF INFECTIOUS DISEASE WITH Ebola and COVID-19, has remained a challenge in Africa. Even now, in the ongoing outbreaks of mpox, Marburg and Sudan Ebola DISEASE
LINE 42: As of January 2025, eight African countries (Egypt, Ghana, Nigeria, Rwanda, Senegal, South Africa, Tanzania, and Zimbabwe) have successfully achieved ML3 status for REGULATION OF ACCESS TO medicines and vaccines….
South Africa and Egypt achieved ML3 for vaccine production…Egypt stands out as the only country in Africa to attain this important milestone for both medicines and vaccines – FOR PRODUCTION?
TABLE – is labelled as ‘Table 3’ - and would benefit from an explanatory legend, making this section a lot clearer.
Author Response
- Add headings and subheadings may give more structure to the document.
Response: Thank you for your suggestion. We have added headings as suggested.
- Use of ‘countermeasure’ throughout – not a term so commonly used in health - replace with 'medical products'.
Response: Thank you for your suggestions. This has been addressed.
Line 12: Africa, with a sizable population, has been affected by recurrent disease outbreaks and yet its health systems have [besides its] limited resilience [of health systems] to mitigate the impacts of these outbreaks.
Response: Thank you. We have now corrected the sentence.
One of the challenges to responding to these outbreaks is the limited access to medical Countermeasures/PRODUCTS such as vaccines, therapeutics and diagnostics.
Response: Thank you. We have now corrected it as medical products.
LINE 15: ..The role of National Regulatory Authorities (NRAs) in contributing to/ENABLING access to SAFE AND EFFECTIVE medical countermeasures/PRODUCTS, including vaccines, is indispensable/CRITICAL. However, most NRAs in Africa lack the required capacities and capabilities to optimally support the SUPPLY AND local manufacturing of ..
Response: Thank you. We have now edited accordingly.
LINE 22: ..during responseS to public health
Response: Thank you. We have now edited accordingly.
LINE 23: To continuously evaluate the level of CAPACITY AND CAPABILITY OF NRAs, the World Health Organization (WHO)
Response: Thank you. We have now edited accordingly.
LINE 32: is a fundamental shift in/REQUIRED FOR the continent's New Public Health Order that
Response: Thank you. We have now edited accordingly.
LINE 34: Notably, the huge disparity in access to medical countermeasures/PRODUCTS such as vaccines during public major public health emergencies, [such] as OCCUR DURING SPREADS OF INFECTIOUS DISEASE WITH Ebola and COVID-19, has remained a challenge in Africa. Even now, in the ongoing outbreaks of mpox, Marburg and Sudan Ebola DISEASE
Response: Thank you. We have now edited accordingly.
TABLE – is labelled as ‘Table 3’ - and would benefit from an explanatory legend, making this section a lot clearer.
Response: Thanks for the suggestion. We have now included a clear label
Round 2
Reviewer 2 Report
Comments and Suggestions for Authors
The authors have incorporated additional information to strengthen the manuscript wherever possible, so I recommend this commentary to the editor for publication.
Author Response
The authors have incorporated additional information to strengthen the manuscript wherever possible, so I recommend this commentary to the editor for publication.
Response: Thank you.
We have also addressed editorial comments (adding an abstract and more words). Thank you.